# Evaluation of Transcatheter Alcohol-Mediated Perivascular Renal Denervation to Treat Resistant Hypertension

**DOI:** 10.3390/jcm9061881

**Published:** 2020-06-16

**Authors:** Adam Janas, Marek Król, Mariusz Hochuł, Monika Jochymczyk, Claudia Hayward-Costa, Helen Parise, Nicole Haratani, Tim Fischell, Wojciech Wojakowski

**Affiliations:** 1Center of Cardiovascular Research and Development, American Heart of Poland, 40-534 Ustroń, Poland; drkrol@gmail.com (M.K.); mhochul@vp.pl (M.H.); monixj87@gmail.com (M.J.); 2Faculty of Medicine and Health Science, Andrzej Frycz-Modrzewski Krakow University, 30-705 Kraków, Poland; 3Ablative Solutions Inc., San Jose, CA 95119, USA; chcosta@ablativesolutions.com (C.H.-C.); hparise@ablativesolutions.com (H.P.); nikkei37@gmail.com (N.H.); taf@ablativesolutions.com (T.F.); 4Division of Cardiology and Structural Heart Diseases, Medical University of Silesia, 40-635 Katowice, Poland; wojtek.wojakowski@gmail.com

**Keywords:** resistant hypertension, renal denervation, local drug delivery

## Abstract

Catheter-based renal denervation (RDN) has been investigated for hypertension (HTN) treatment with variable success. One of the novel approaches to RDN is the delivery of micro-doses of dehydrated alcohol to the adventitial space of the renal artery to perform perivascular ablation of the sympathetic nerves. We sought to assess the safety and efficiency of transcatheter alcohol-mediated perivascular renal denervation in patients with resistant hypertension. Fifty adult patients who had been referred for resistant HTN were screened. To qualify for the study, the patients had to have a mean 24 h systolic pressure ≥ 135 mmHg based upon ambulatory blood pressure monitoring (ABPM) and acceptable renal artery anatomy confirmed by the contrast computer tomography (AngioCT) and nephrologist consultation. Ten patients were eligible for chemical RND. There were no safety issues throughout the 24 months of follow-ups. The mean decrease in the office BP (OBP) was significant during 24 months of follow-up (*p* < 0.01). The difference in the BP in the ABPM was statistically significant in the 1st, 3rd and 12th months (*p* < 0.01), whereas during the 3-month follow-up, a trend was observed. The modifications of anti-hypertension drugs throughout the follow-up period were minimal. This study has shown that transcatheter alcohol-mediated renal denervation in patients with resistant hypertension is feasible and safe. Nevertheless, it is a hypothesis-generating study.

## 1. Introduction

Hypertension (HTN) is the leading treatable risk factor for the cardiovascular disease [1,2,3]. The number of patients with HTN is growing [4]. It is estimated that 1.5 billion people will suffer from hypertension in 2025 [5]. Nonetheless, the number of patients with blood pressure (BP) values controlled to guideline-recommended target values remains low [6]. The rationale for an endovascular approach to treat hypertension is based on the theory of sympathetic crosstalk between the kidneys and the brain [7]. Moreover, renal sympathetic nerves are involved in the development and maintenance of hypertension [8,9]. Catheter-based renal denervation (RDN) has been under investigation for hypertension treatment with variable success [10]. The first systems for RDN employed the radiofrequency energy. More recently, systems utilizing ultrasound and chemical denature have been developed [11,12,13]. One approach has been to use the Peregrine Catheter, a specialized vascular drug-delivery device used to deliver micro-doses of dehydrated alcohol, a potent neurolytic agent, locally into the adventitial space of the renal artery to perform a perivascular ablation of the afferent and efferent sympathetic nerve bundles [14,15].

Therefore, we sought to assess the safety and efficiency of transcatheter alcohol-mediated perivascular renal denervation in patients with resistant hypertension.

## 2. Materials and Methods

### 2.1. Study Population

Fifty adult patients who had been referred for resistant HTN were screened after signing informed consent forms. To qualify, all subjects had to experience resistant hypertension defined as an office BP (OBP) > 160 mmHg (>150 mm Hg if the subject had type 2 diabetes) while taking at least 3 antihypertensive medications, including a diuretic agent for at least 4 weeks. Moreover, the patients had to have mean 24 h systolic pressure ≥ 135 mmHg based upon ambulatory blood pressure monitoring (ABPM). In addition, there had to be acceptable renal artery anatomy confirmed by AngioCT and nephrologist consultation. The inclusion and exclusion criteria are outlined in Table 1 and Table 2, respectively.

### 2.2. Study Design

This was a proof-of-concept, prospective, open-label, single arm study, to investigate the safety and efficacy of alcohol-mediated renal sympathetic denervation, using a novel 3 needle-based delivery device (Peregrine System™ Infusion Catheter, Ablative Solutions, Inc., San Jose, CA, USA) to treat patients with resistant hypertension. The detailed device description has been previously published [14]. Micro doses of dehydrated alcohol (0.3 mL per renal artery) were used as the neurolytic agent.

The primary safety endpoints included procedure and device-related adverse events and any post-procedure adverse cardiovascular or vascular complications, deterioration in renal function, and renal artery abnormality. The primary efficacy endpoint included a change in the OBP and the ABPM at each time point.

The study was approved by each institution’s Research Ethics Committee and is registered with Clinical Trials Registry (Registration No. NCT02155790). The trial was sponsored by Ablative Solutions, Inc., San Jose, CA, USA.

### 2.3. Study Procedures

#### 2.3.1. Baseline

Following written informed consent, medical histories and physical examinations including the OBP measurement were completed. The OBP was collected according to the ESC/ESH guidelines [16]. Each enrolled patient was provided with an Automatic BP Monitor for collection of the office and home BP. The anti-hypertensive medication regimen was reviewed and was required to be stable and unchanged for a minimum of 4 weeks. The 24 h ambulatory BP was obtained using an Ambulatory Blood Pressure System. The ABPM was valid if at least 90% of the planned measurements were taken. Computed Tomography Angiography of the renal arteries was performed. After 30 days of the screening period, the blood and urine were collected for determining the blood count, basic metabolic profile, serum creatinine, estimated glomerular filtration rate (eGFR) and blood urea nitrogen. Prior to treatment, and after the 30-day run-in period, the OBP was measured, a 12-lead ECG was recorded, and the medication and BP logs were reviewed.

#### 2.3.2. Renal Denervation Procedure

The renal artery was engaged using a 7-F guiding catheter, introduced via the femoral artery under fluoroscopic guidance. Intravenous unfractionated heparin was given to attain an activated clotting time of >250 s. No general anesthesia was used in any of the subjects. Analgesics were used only if the patient experienced significant discomfort.

Angiography of the renal arteries was performed before the intervention. After the initial angiography, the Peregrine Catheter was advanced through the guiding catheter and into the distal half of the renal artery, typically just past the midpoint.

The Peregrine Catheter consists of a catheter with three self-centering guide tubes separated at 120 degrees from one another that, once deployed by a simple mechanism via the catheter handle, allow the distal end of each tube to establish contact with the renal arterial wall directing the center of the device to assume a position at or close to the middle of the renal artery lumen. Once positioned in the desired landing zone, three radio-opaque microneedles (0.008 inch/220 mm) were advanced via the guide tubes (Figure 1) through the renal artery wall (penetrating the arterial wall by 3.5 ± 0.25 mm, the approximate distance between the lumen–intima interface and the transition between the adventitia and surrounding tissue) into the adventitia and surrounding perivascular space. The neurolytic agent (0.3 mL of dehydrated alcohol injection, USP Akorn, Inc., Lake Forest, IL, USA) was then delivered slowly using a 1 mL Luer-lock syringe, connected at an infusion port located at the proximal end of the control handle. Once the alcohol was infused, the dead space within the catheter was flushed with 0.1 mL of normal saline. Angiography was performed after the withdrawal of the Peregrine Catheter.

As a safety measure per protocol, in the first 5 subjects, only 1 renal artery was treated (unilateral denervation) during the initial procedure. These first 5 subjects returned 4 weeks after the initial treatment for angiography of the treated renal vessel. Once angiography confirmed no issues in the previously treated renal artery (5/5), the treatment of the contralateral renal artery was performed to complete the bilateral renal denervation. The subsequently treated 5 subjects underwent bilateral denervation during a single procedure. The detailed procedure descriptions had been previously published [14].

#### 2.3.3. Pre-Discharge 

The serum creatinine estimated glomerular filtration rate (eGFR) and blood count were assessed before the discharge. Patients were discharged from the hospital on the following day after the procedure if there were no issues with the laboratory measurements or the access site.

#### 2.3.4. Follow-Up

After discharge, each patient had a follow-up office visit with the following data collection at each designated time period (7 days and 1, 3, 6, 12 and 24 months): the office BP and ABPM assessment, review of medication for changes, 12-lead ECG and blood and urine collection. A Computed Tomography Angiography (CTA) was performed at 1, 6 and 24 months after the procedure. The study flow chart with the patient disposition is on Figure 2.

### 2.4. Study Oversight

All Serious Adverse Events (SAE) were assessed by an Independent Medical Reviewer (IMR). Monitoring and data analysis were performed by Ablative Solutions, Inc. The angiography and CTAs were assessed by an independent core lab. The corresponding author had full access to the study data.

## 3. Statistical Analysis

This was a small cohort, proof-of-concept trial. There were no powered endpoints in the study. The statistical analysis was performed as intention-to-treat. The continuous variables are presented as mean ± SD or median (IQR). The normality of data was verified with the use of box plots and the Kolmogorov–Smirnov normality test. For normally distributed data, comparisons of primary and secondary outcomes between time points were analyzed using paired t-tests. In cases where the data were not normally distributed, the non-parametric Wilcoxon signed-rank test was used to analyze the data. All categorical parameters were summarized using frequencies and percentages. GraphPad 6 Prism (GraphPad Software, San Diego, CA, USA) was used for statistical analysis.

## 4. Results

Between July 2014 and May 2015, fifty patients were screened and enrolled. Ten patients met the inclusion/exclusion criteria and were treated. The baseline clinical characteristics are shown in Table 3. In all patients, selective renal angiography was performed, and angiographic documentation of the catheter position was documented. During the procedure, a mean of 80 ± 36 mL of contrast medium was used. The procedural characteristics are summarized in Table 4. There was a 100% success rate in the 15 procedures (10 unilateral denervation procedures and 5 bilateral procedures) with no device failures or deficiencies reported. In each case, the planned alcohol volume (0.3 mL/artery) was delivered successfully.

### 4.1. Safety Objectives

There were no primary safety issues during the procedure or throughout the follow-up. Moreover, control angiography revealed no cases of artery spasm after the procedure. There was no deterioration in renal function during the follow-up (Table 5 and Figure 3). There were no significant differences between the follow-up time points. Three serious adverse events (SAEs) occurred in two subjects. One patient had what was described as an inflammation of the duodenum mucosae membrane, subsequently confirmed in gastroscopy, approximately 6 months after the procedure. The SAE resolved without sequelae and was determined by the IMR as not related to the study device or study procedure. SAE number 2 and 3 occurred in the same patient and were described as upper respiratory tract infections treated with antibiotics and subsequent diabetes intensification. Both SAEs were resolved without sequelae and were determined by the IMR as not related to the study device and study procedures. During two procedures, patients felt lower back pain during and just after alcohol injection described as 2 points on a 10 point scale. Both patients received, intravenously, 500 mg of paracetamolum, which relieved the discomfort.

### 4.2. Efficiency Objectives

Compared to the baseline, the OBP of the entire cohort decreased at all time points (*p* < 0.001). The average OBP (mmHg) at the baseline was 168 ± 8 (mmHg). The resulting average OBP (mmHg) reduction from 1 month, 3 months, 6 months, 12 months and 24 months was −44 ± 11, −29 ± 13, −37 ± 14, −28 ± 14, −21 ± 13 and −25 ± 7, respectively (Figure 4). Over the follow up period, 60% of treated patients had a reduction in systolic OBP by more than 10% in comparison to the baseline.

The average 24 h ambulatory BP at baseline was 146 ± 12 mmHg. The average 24 h ABP (mmHg) reduction from the baseline at 1, 3, 6 (within window), 6 (outside window), 12 and 24 months was −12 ± 6, −7 ± 10, −5 ± 4 (within window *n* = 6), −3 ± 10 (outside window *n* = 10), −6 ± 5 and −1 ± 6 respectively. The difference in the systolic BP in the ABPM was statistically significant at 1, 3 and 12 months (*p* < 0.01), whereas during 3 months of follow-up, the trend was seen. At 6 and 24 months of follow-up, there were no significant differences (Figure 5). One patient was lost to follow-up, and the 24 months visit was not performed.

The reduction in the OBP and the ABPM were achieved with minimal modifications of anti-hypertensive drugs throughout the follow-up period. There were no changes in antihypertensive medications reported at either 12 months or 24 months; 6 subjects (67%) had no changes in their antihypertensive regimen. At 24 months, one subject (11%) had the number of antihypertensive medications increased by one; one subject had the number of antihypertensive drugs increased by three and one subject had the number of antihypertensive medications decreased by one.

## 5. Discussion

In patients with resistant hypertension on multiple medications, a bilateral infusion of 0.3 mL of alcohol in the perivascular space of the renal arteries using the Peregrine Catheter has shown an excellent safety profile, a relatively short time frame and a nearly painless manner of procedure. Moreover, this study has also shown a clinically meaningful reduction in the OBP and the ABPM.

The renal sympathetic nerves are located in the adventitia surrounding the renal arteries. Historically, RDN was originally described using radiofrequency (RF) [17,18,19]. Given some early challenges with the completeness of denervation related to nerve distribution and penetration depth using the early generation RF system, a revised approach of endovascular RDN for resistant hypertension was proposed and subsequently investigated in clinical trials [20,21]. The randomized, sham-controlled SPYRAL HTN-OFF and SPYRAL HTN-ON have confirmed the potential of this approach in patients with and without antihypertensive medications [11,12]. Moreover, in the RADIANCE- HTN SOLO, ultrasound energy thermal ablation also appeared to be a promising method of RDN for BP reduction [13].

The Peregrine Catheter utilizes a novel design to allow “chemical renal denervation,” using micro-dosing of a neurolytic agent delivered to the periadventitial area around the renal arteries. The Peregrine Catheter was used under fluoroscopic guidance to place guide tubes, and subsequently, three micro-needles in the renal artery. In this study, dehydrated alcohol was used as the neurolytic agent, and was infused through the needles directly into the perivascular space. The preclinical evaluation of this approach suggested that this may provide a safe and reproductible means of achieving deep and circumferential sympathetic nerve disruption by denaturation [15]. In this study, by Fischell et al. [15], there was substantial sympathetic nerve disruption, without histological or angiographic evidence of dissection, perforation, necrosis or other significant disruption of the normal constituents of the renal artery. Moreover, histopathology demonstrated circumferential treatment zones of neurolysis at depths of up to 16 mm from the intima.

Another important observation was the apparent safety of this means of renal denervation. There were no observed periprocedural or long-term complications. Moreover, the creatinine ratio was stable during the follow-up. In addition, the procedural aspects of chemical renal denervation with the Peregrine Catheter are distinguished from “energy-based” thermal ablation techniques using radiofrequency or ultrasound energy. The Peregrine Catheter does not require a capital equipment energy generator-console. This reduced the complexity of set-up and costs associated with the procedure. The single alcohol injection in the main renal artery shortened the time and may have reduced contrast volume. The mean (±SD) volume of the contrast used during this trial (40 ± 18 mL/artery) was lower than in any other recently reported trial (RADIANCE-HTN SOLO: 141 ± 69 mL vs. SPYRAL HTN-ON MED: 270.8 ± 101.6 mL vs. SPYRAL HTN-OFF MED: 251 ± 99.4 mL) (11–13). Shortly after the injection, alcohol anaesthetizes the afferent, sensory nerves, which is typically the cause of pain during an RDN procedure. In this study, no general anesthesia was required on any of the patients treated, with an analgesic administered only in the case of minimal discomfort.

## 6. Limitations

The main limitation of the study was the small sample size. This study was designed as a proof-of-concept-study and was not meant to be a definitive or pivotal study of this technology. However, it was encouraging to observe some blood pressure reduction at several time points during the follow-up. Moreover, the efficiency endpoint was not evaluated by the measurement of endocrinal secretion i.e., noradrenaline. Due to the proof-of-concept design of this study, the sham arm was not present.

## 7. Conclusions

This study has shown that transcatheter alcohol-mediated perivascular renal denervation in patients with resistant hypertension is feasible and appears to be safe. The study is hypothesis-generating but will require additional studies in subjects both off medications and on medications, with appropriate controls to ultimately determine the role of this therapy in the management of patients with hypertension. The randomized, blinded, sham-controlled Target BP OFF-Med (European study, phase 2) and Target BP-I on meds (Global study, phase 3) studies are ongoing, and will further evaluate the safety and efficacy of this novel approach to renal denervation.

## Figures and Tables

**Figure 1 jcm-09-01881-f001:**
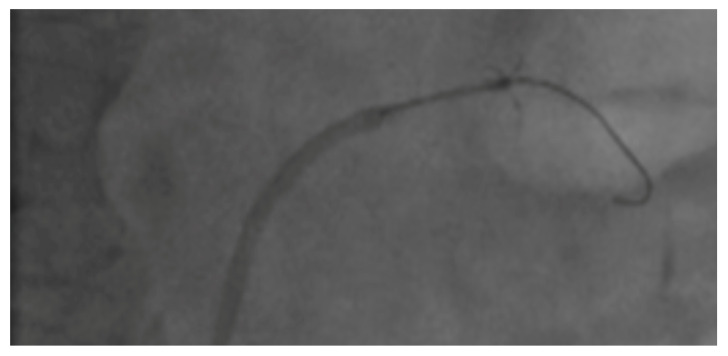
The Peregrine Catheter in the renal artery with open tubes and exerted needles.

**Figure 2 jcm-09-01881-f002:**
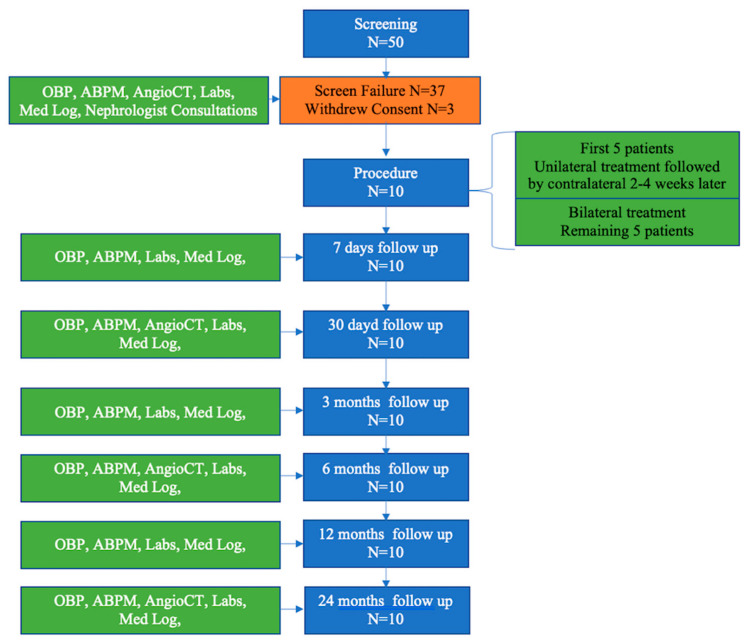
Study flow chart with subject disposition. Office Blood Pressure (OBP), Ambulatory Blood Pressure (ABPM), Contrast Computer Tomography (Angio CT).

**Figure 3 jcm-09-01881-f003:**
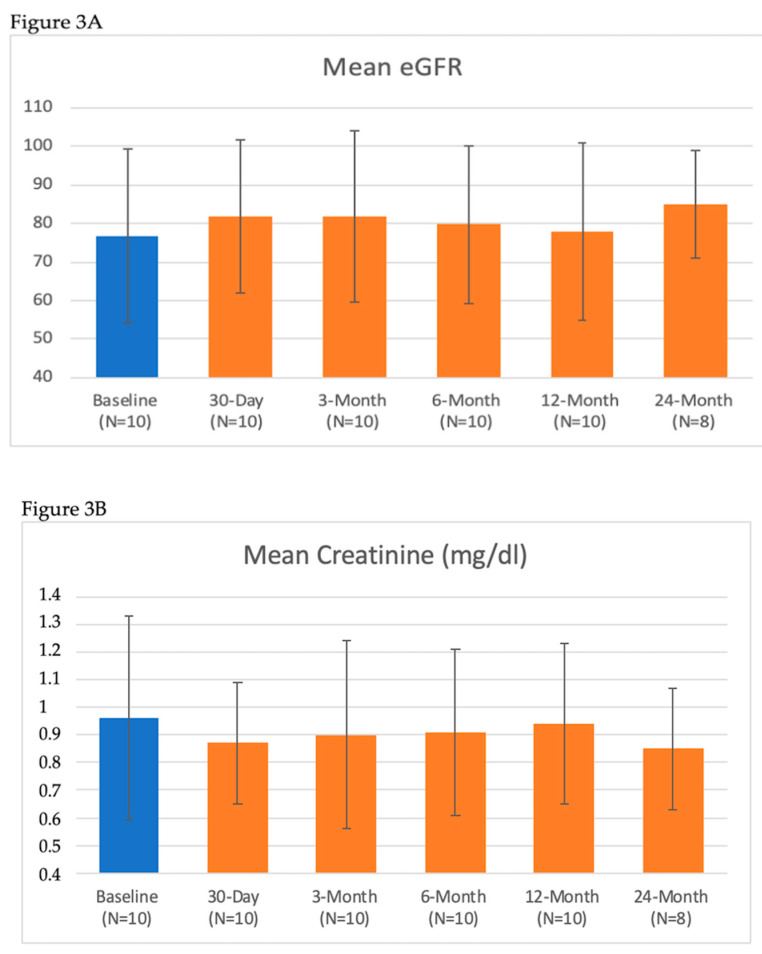
Renal function: (**A**) Mean Estimated Glomerular Filtration Rate (eGFR), (**B**) Mean Serum Creatinine, (**C**) Mean Blood Urea Nitrogen (BUN).

**Figure 4 jcm-09-01881-f004:**
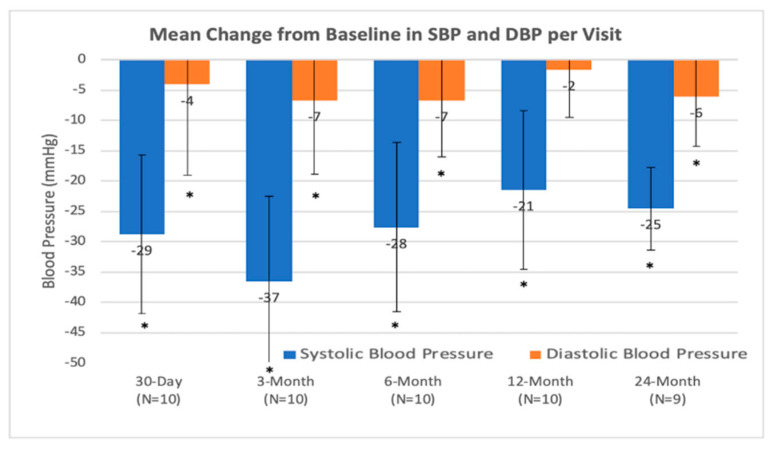
Mean change from baseline in systolic blood pressure and diastolic blood pressure per visit in office blood pressure. * statistically significant.

**Figure 5 jcm-09-01881-f005:**
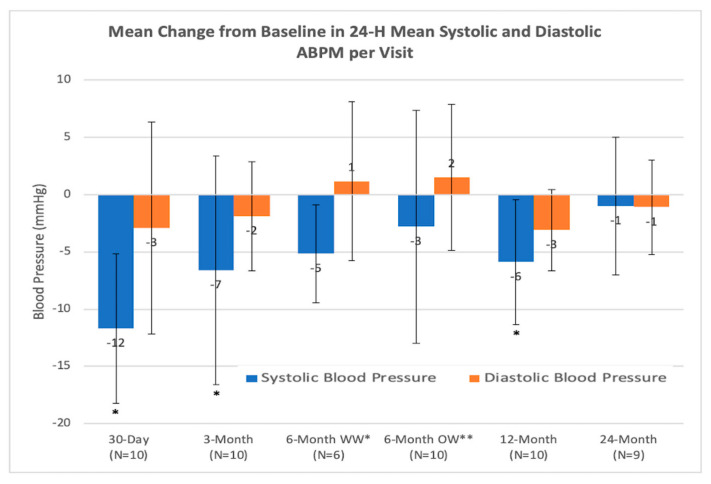
Mean change from baseline in systolic blood pressure and diastolic blood pressure per visit in 24 h ambulatory blood pressure monitoring. * statistically significant; ** outside the follow up window.

**Table 1 jcm-09-01881-t001:** Inclusion criteria.

Inclusion Criteria
Adult patient, aged 18–75, male or female
Patient has a clinic systolic blood pressure ≥ 160 mmHg (or ≥150 mmHg in type 2 diabetic patients) based on the average of 3 office/clinic measurements taken manually
Patient has a daytime mean systolic pressure ≥ 135 mmHg based on 24 h ambulatory blood pressure monitoring
Patient is receiving a stable medication regimen of at least 3 anti-hypertensive medications of different classes (for at least 4 weeks), one of which must be a diuretic, and the medication regimen is not expected to change for at least 1 month
Patient has an eGFR ≥ 45 mL/min, based on the Chronic Kidney Disease Epidemiology Colaboration (CKD-EPI) equation
Patient has optimal renal artery anatomy (no clear abnormalities) based on Investigator’s evaluation of computed tomography examination and/or angiogram including: Single artery of 5–7 mm in diameter (two arteries are acceptable if diameter of second artery is ≤2 mm) No aneurysms No excessive tortuosity
No previous stenting or balloon angioplasty of the renal arteries
Patient has provided written informed consent

**Table 2 jcm-09-01881-t002:** Exclusion criteria.

Exclusion Criteria
Patient has known or suspected secondary hypertension
Patient has type 1 diabetes mellitus
Patient requires chronic oxygen support
Patient has primary or secondary pulmonary hypertension
Patient has a known bleeding diathesis or is receiving anticoagulant drugs during the 7 days prior to treatment
Patient has thrombocytopenia (platelet count <100,000 platelets/µL)
Patient is pregnant or nursing
Patient has significant imaging-assessed renovascular abnormalities including short length main renal artery and renal artery stenosis > 70% of the normal diameter segment
Patient has history of nephrectomy, kidney tumor or hydronephrosis
Patient is known to have a unilateral non-functioning kidney or unequal renal size (>2 cm difference in renal length between kidneys)
Patient has a renal transplant
Patient has a history of kidney stones
Patient has a history of heterogeneities in the kidney such as cysts or tumors
Patient has a history of pyelonephritis
Patient has a history of myocardial infarction, unstable angina pectoris, or cerebrovascular accident within the last six months
Patient has hemodynamically significant valvular heart disease
Patient has heart failure (NYHA III or IV) or had an ejection fraction ≤ 30%
Patient has a known allergy to contrast media
Patient has a life expectancy of <12 months
Patient is currently enrolled in other potentially confounding research, i.e., another therapeutic or interventional research trial. Patients enrolled in observational registries may still be eligible

**Table 3 jcm-09-01881-t003:** Demographics and pharmacotherapy.

Characteristic	All Treated Subjects *n* = 10
Age, years (mean (min; max))	59.9 (46;67)
Male gender	50% (5/10)
Renal Disease	0% (0/10)
Chronic Kidney Disease	10% (1/10)
Hypertension	100% (10/10)
Diabetes (Type II)	40% (4/10)
Arrhythmia	0% (0/10)
Atrial Fibrillation/Atrial Flutter	0% (0/10)
Prior Stroke or TIA	10% (1/10)
Peripheral Vascular Occlusive Disease	10% (1/10)
Hyperlipidemia/Dyslipidemia	60% (6/10)
Angio-determined Coronary Artery Disease	30% (3/10)
Prior Myocardial Infarction	20% (2/10)
Previous History of Congestive Heart Failure	0% (0/10)
Chronic Obstructive Pulmonary Disease	0% (0/10)
History of Smoking	30% (3/10)
Defined Daily Dose (DDD) of anti-hypertensive	4.8 ± 1.3
Thiazide Diuretic	90% (9/10)
Loop Diuretics	30% (3/10)
Beta-Blocker	90% (9/10)
Alpha -1 Blocker	50% (5/10)
Centrally Acting Alpha Agonist	20% (2/10)
Direct Acting Vasodilator	50% (5/10)
ACE Inhibitor	70% (7/10)
Calcium Channel Blocker	50% (5/10)
Angiotensin II Receptor Blocker	50% (5/10)
Diuretic Aldosterone Receptor Blockers	20% (2/10)

**Table 4 jcm-09-01881-t004:** Procedural parameters.

Parameters	Number of Procedures (*n* = 15 Procedures) (*n* = 20 Arteries) Mean ± SD (Min; Max)
Time from Insertion of Peregrine Catheter into Renal Artery to Removal from Introducer Sheath, per artery (min)	8 ± 5 (4; 22)
Distance from Ostium to First Bifurcation–Right (mm)	47.7 ± 10.4 (32.0; 67.0)
Distance from Ostium to First Bifurcation–Left (mm)	33.5 ± 12.0 (12.1; 51.0)
Distance to Ostium from Infusion Site–Right (mm)	25.8 ± 7.3 (9.4; 35.0)
Distance to Ostium from Infusion Site–Left (mm)	16.3 ± 6.8 (8.2; 29.9)
Volume of Alcohol Infused, per artery (mL)	0.3
Total Volume Contrast Used, per artery (mL)	40 ± 18 (20; 100)
Total IV Fluid Used, per artery (mL)	1003 ± 507 (0; 2058)
Fluoroscopic Time, per artery (min)	6.0 ± 5.4 (2.0; 26.0)
Average Hospital Stay (Days)—Median (IQR)	2 (2; 3)

**Table 5 jcm-09-01881-t005:** Mean BUN, mean serum creatinine and eGFR during follow up.

Renal Lab Parameter	Baseline (Mean ± SD) *n* = 10	7-Day (Mean ± SD) *n* = 10	30-Day (Mean ± SD) *n* = 10	3-Month (Mean ± SD) *n* = 10	6-Month (Mean ± SD) *n* = 10	12-Month (Mean ± SD) *n* = 10	24-Month (Mean ± SD) *n* = 8
BUN (mg/dL)	17.4 ± 5.3	17.5 ± 3.8	15.7 ± 3.4	16.5 ± 3.2	17.4 ± 4.5	18.3 ± 4.0	16.2 ± 4.1
Serum Creatinine (mg/dL)	0.96 ± 0.37	0.96 ± 0.37	0.87 ± 0.22	0.90 ± 0.34	0.91 ± 0.30	0.94 ± 0.29	0.85 ± 0.22
eGFR (mL/min/1.73 m^2^)	78 ± 21	76 ± 21	82 ± 20	82 ± 22	80 ± 20	77 ± 20	85 ± 14
≥25% Reduction in eGFR		10% (1/10)	0% (0/10)	0% (0/10)	0% (0/10)	10% (1/10)	0% (0/8)

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
