# Peer review of "Evaluation of Transcatheter Alcohol-Mediated Perivascular Renal Denervation to Treat Resistant Hypertension"

_jcm, 2020, doi:10.3390/jcm9061881_

Round 1

Reviewer 1 Report

Results

Figure 3: what is the difference between mean eGFR Figure 3A and Figure 3C. To me they appear to be the same figure?

Did the authors determine efficacy of denervation in these patients (ie noradrenaline?)

In the limitations section, the authors could add that the study does not have a sham arm and perhaps provide recommendation of what the sham procedure may entail if this was to be followed up as a larger scale study.

Author Response

Reviewer 1

Results

1) Figure 3: what is the difference between mean eGFR Figure 3A and Figure 3C. To me they appear to be the same figure?

Ad 1) Figure 3 was corrected; the mean serum creatinine figure was added instead of doubled mean eGFR figure.

2) Did the authors determine efficacy of denervation in these patients (ie noradrenaline?)

Ad 2) No, the efficiency of denervation measured by endocrinal secretion (NE spillover) was not measured. This was acknowledged in the limitations. P13 L:283-284

 3) In the limitations section, the authors could add that the study does not have a sham arm and perhaps provide recommendation of what the sham procedure may entail if this was to be followed up as a larger scale study.

 Ad 3) The limitation section is improved according to your suggestion. P13 L:283-284

Reviewer 2 Report

The authors of the manuscript "Evaluation of a Transcatheter Alcohol-Mediated Perivascular Renal Denervation to Treat Resistant Hypertension" describe a pilot study, that they have performed several years ago. Although the paper describes an important topic and novel technology, I have some concerns (and minor comments):

Major:

  • In general the manuscript is written in a seemingly sloppy manner; the level of English is highly variable, the use of abbreviations is strange (for instance already in the abstract: RDN-RND, AngioCT does not return in the abstract), etc.
  • The researchers aimed to assess the efficiency (next to the safety) of transcatheter alcohol-mediated perivascular renal denervation. However, the study is a very limited first-in-man study and should be described as such. Trying to show a treatment effect in a non-controlled open-label study in only 10 patients should be avoided. Even more, the data are >5 years old. How come that a FIM study like this is still unpublished? Therefore, throughout the manuscript major textual changes should be made to meet to the character of the study. 
  • Even more, the authors show a 'treatment effect' on office BP; however ABPM shows no effect on the long run. Minimize these data in the manuscript and put more emphasis on the safety profile of alcohol ablation.
  • I miss the data on the painfulness of this technique. In the discussion section (page 10) the authors describe it to be a "nearly painless procedure". However, no data are presented in the results section.
  • The changes in medication has been described in a qualitative manner (Page 10; lines 221-226). I would advise to describe the prescribed dosages of antihypertensive drugs converted to daily defined doses using conversion factors as provided by the World Health Organization Drug Classification (http://www.whocc.no/atcddd/). Using daily defined doses and total prescribed dosages, daily use (DU) of all antihypertensive drugs can be calculated.
  • The adverse events are described. However, blood and urine parameters, particularly concerning kidney function on all time points described in the methods section should be described in a Table; again; safety is the main point of this study.

Minor:

  • Figure 1 is not very clear; I can not distinguish the needles on the image.
  • In total, 50 patients were 'enrolled' as described in the manuscript. However; 50 patients were screened and 80% of patients were excluded for participation in the study. Please change throughout the manuscript.
  • In the results section (page 8/9) 3A and 3C show the same table (eGFR); sloppy error or did the authors want to show a different parameter in 3C?
  • Figure 4 shows changes in systolic BP and diastolic BP; however, the bars start at +20 mmHg?? Even more, I presume that the blue bars are SBP and orange is DBP; however not described.

Author Response

Reviewer 2

 The authors of the manuscript "Evaluation of a Transcatheter Alcohol-Mediated Perivascular Renal Denervation to Treat Resistant Hypertension" describe a pilot study, that they have performed several years ago. Although the paper describes an important topic and novel technology, I have some concerns (and minor comments):

Major:

1) In general, the manuscript is written in a seemingly sloppy manner; the level of English is highly variable, the use of abbreviations is strange (for instance already in the abstract: RDN-RND, AngioCT does not return in the abstract), etc.

Ad 1) The manuscript was corrected by professional English editor specialized in medical manuscripts    editing. I hope those corrections have improved the quality of the mansucript.

2) The researchers aimed to assess the efficiency (next to the safety) of transcatheter alcohol-mediated perivascular renal denervation. However, the study is a very limited first-in-man study and should be described as such. Trying to show a treatment effect in a non-controlled open-label study in only 10 patients should be avoided. Even more, the data are >5 years old. How come that a FIM study like this is still unpublished? Therefore, throughout the manuscript major textual changes should be made to meet to the character of the study. 

Ad 2) This was not FIM study (FIM already published by Fishell et al.) but as described in the manuscript it was proof-of-concept study.  This study was designed with emphasis for safety endpoints including both peri procedural as well as long term follow up (systematic follow up with angio CT). Safety assessment included clinical, laboratory and imaging long term follow up. Each procedure was described in the method and subsequently in the results paragraph. Nevertheless, the efficiency must be present in a prof-of-concept trials to assure safety and risk – benefit profile. The efficacy assessment was prespecified at the time of the study design, so it has to be reported according to common standards. The efficacy evaluation was necessary also because of the futility signals from HTN-3 study published during the development of the Peregrine technology. We do not claim the study proved RND with alcohol to effective, but we found reassuring hypothesis-generating data to start the randomized study. 

3)Even more, the authors show a 'treatment effect' on office BP; however, ABPM shows no effect on the long run. Minimize these data in the manuscript and put more emphasis on the safety profile of alcohol ablation.

Ad 3) The ABPM data was minimalized.    P:11 L:218-220

 4) I miss the data on the painfulness of this technique. In the discussion section (page 10) the authors describe it to be a "nearly painless procedure". However, no data are presented in the results section.

Ad 4) This data has been completed in results section. P:8 L:190-192

5) The changes in medication has been described in a qualitative manner (Page 10; lines 221-226). I would advise to describe the prescribed dosages of antihypertensive drugs converted to daily defined doses using conversion factors as provided by the World Health Organization Drug Classification (http://www.whocc.no/atcddd/). Using daily defined doses and total prescribed dosages, daily use (DU) of all antihypertensive drugs can be calculated.

Ad 5) The dosage of antihypertensive drugs was changed. Table 3.

 6) The adverse events are described. However, blood and urine parameters, particularly concerning kidney function on all time points described in the methods section should be described in a Table; again; safety is the main point of this study.

Ad 6) The table have been added. Table 5.

Minor:

1) Figure 1 is not very clear; I cannot distinguish the needles on the image.

Ad 1) Figure A with open tubes was deleted. Picture with open tubes and needles is presented in the manuscript.

2)In total, 50 patients were 'enrolled' as described in the manuscript. However; 50 patients were screened and 80% of patients were excluded for participation in the study. Please change throughout the manuscript.

Ad 2) The change was performed according to Reviewer 2 suggestions. P: 1 L:17, P2 L: 46

3) In the results section (page 8/9) 3A and 3C show the same table (eGFR); sloppy error or did the authors want to show a different parameter in 3C?

Ad 3) The doubled mean eGFR figure was changed to mean creatinine figure.

4) Figure 4 shows changes in systolic BP and diastolic BP; however, the bars start at +20 mmHg?? Even more, I presume that the blue bars are SBP and orange is DBP; however not described.

Ad 4) The figure was corrected according to Reviewer 2 suggestions.

Thank you for such detailed analysis of the manuscript, it would improve the quality significantly. We hope the changes will satisfy you.

Round 2

Reviewer 2 Report

The manuscript has improved clearly. However, still some modifications need to be made.

The authors conclude in the first paragraph of the discussion section:

"In patients with resistant hypertension on multiple medications, bilateral infusion of 0.3 ml of alcohol in the perivascular space of the renal arteries using the Peregrine Catheter was associated with a clinically meaningful reduction in office and ambulatory BP. The procedural safety profile was excellent, and with a relatively short, and nearly painless procedure in the 15 procedures performed."

In the light of the character of the study these sentences should be interchanged; the first conclusion concerns safety / feasibility. Second concerns efficacy.

Despite the grammar correction some obvious errors are still present (eg: line 192: relived -> relieved; Figures 4 and 5: disatolic -> diastolic). Line 284: "Duo to proof-of-concept designee of this study the sham arm was not..." Duo -> Due. Sentence is not finished...

Please check the manuscript once more in detail.

Author Response

The authors conclude in the first paragraph of the discussion section:

"In patients with resistant hypertension on multiple medications, bilateral infusion of 0.3 ml of alcohol in the perivascular space of the renal arteries using the Peregrine Catheter was associated with a clinically meaningful reduction in office and ambulatory BP. The procedural safety profile was excellent, and with a relatively short, and nearly painless procedure in the 15 procedures performed."

 This sentence was reedited P:12 L:244-247

Despite the grammar correction some obvious errors are still present (eg: line 192: relived -> relieved; Figures 4 and 5: disatolic -> diastolic). Line 284: "Duo to proof-of-concept designee of this study the sham arm was not..." Duo -> Due. Sentence is not finished...

Please check the manuscript once more in detail.

The manuscript was checked and corrected.